# Lectin and *E. coli* Binding to Carbohydrate-Functionalized Oligo(ethylene glycol)-Based Microgels: Effect of Elastic Modulus, Crosslinker and Carbohydrate Density

**DOI:** 10.3390/molecules26020263

**Published:** 2021-01-07

**Authors:** Fabian Schröer, Tanja J. Paul, Dimitri Wilms, Torben H. Saatkamp, Nicholas Jäck, Janita Müller, Alexander K. Strzelczyk, Stephan Schmidt

**Affiliations:** Institute for Organic and Macromolecular Chemistry, Heinrich-Heine-University, Universitätsstr 1, 40225 Düsseldorf, Germany; Fabian.Schroeer@hhu.de (F.S.); tanja.paul@hhu.de (T.J.P.); dimitri.wilms@hhu.de (D.W.); Torben.Saatkamp@uni-duesseldorf.de (T.H.S.); Nicholas.Jaeck@uni-duesseldorf.de (N.J.); jamue112@hhu.de (J.M.); alexander.strzelczyk@hhu.de (A.K.S.)

**Keywords:** P(MEO_2_MA-*co*-OEGMA), PNIPAM, LCST, stimuli responsive polymers, FimH, glycans

## Abstract

The synthesis of carbohydrate-functionalized biocompatible poly(oligo(ethylene glycol) methacrylate microgels and the analysis of the specific binding to concanavalin A (ConA) and *Escherichia coli* (*E. coli*) is shown. By using different crosslinkers, the microgels’ size, density and elastic modulus were varied. Given similar mannose (Man) functionalization degrees, the softer microgels show increased ConA uptake, possibly due to increased ConA diffusion in the less dense microgel network. Furthermore, although the microgels did not form clusters with *E. coli* in solution, surfaces coated with mannose-functionalized microgels are shown to bind the bacteria whereas galactose (Gal) and unfunctionalized microgels show no binding. While ConA binding depends on the overall microgels’ density and Man functionalization degree, *E. coli* binding to microgels’ surfaces appears to be largely unresponsive to changes of these parameters, indicating a rather promiscuous surface recognition and sufficiently strong anchoring to few surface-exposed Man units. Overall, these results indicate that carbohydrate-functionalized biocompatible oligo(ethylene glycol)-based microgels are able to immobilize carbohydrate binding pathogens specifically and that the binding of free lectins can be controlled by the network density.

## 1. Introduction

The interactions between glycans at the cell surface and carbohydrate-binding receptors control many biological functions on the cellular scale, e.g., cell adhesion, communication, signal transduction or fertilization [1,2]. The invasion of bacterial pathogens and their colonization in tissues is often controlled by carbohydrate interactions as well [3,4]. Elongated hair-like protein complexes called pili or fimbriae present lectins that bind to the cells’ glycocalyx as a first step of pathogen invasion and the development of infectious diseases. A well-studied example of a carbohydrate-binding pathogen is *E. coli* [5,6]. Via the lectin FimH at the tip of the fimbriae, *E. coli* binds to mannoside units at the cell surface causing a range of infectious diseases, e.g., highly fatal neonatal meningitis [7,8,9] or urinary tract infections [10]. As a result of the ongoing antibiotic resistance crisis, such infections have become difficult to treat [11,12]. This is caused by antibiotics that kill the bacteria, e.g., by the inhibition of protein synthesis or interrupting their cell division process, thereby causing evolutionary pressure leading to antibiotic resistance [13,14]. A new class of carbohydrate-based drugs inhibiting the anchorage of the pathogens to the glycocalyx have been shown to prevent and cure infections without inducing antibiotic resistance [15,16,17,18,19]. Potent inhibitors of carbohydrate-mediated pathogen adhesion are usually based on multivalent macromolecular glycoconjugates presenting many carbohydrate subunits, thus binding to bacterial lectins with high avidity [20,21,22,23,24,25].

Microgel scaffolds have recently been developed as a platform to study carbohydrate–lectin interactions and to capture carbohydrate-binding bacteria [26,27,28,29,30]. In addition to their straightforward synthesis, microgels allow for preparing robust surface coatings through simple physisorption methods, e.g., via drop-casting, spin coating or dip-coating [31] Carbohydrate-functionalized microgels are highly hydrated and soft, thus mimicking properties of the extracellular matrix or glycocalyx, which sets them apart from other glycan-presenting scaffolds. For synthesizing microgel scaffolds, polymers with a lower critical solution temperature (LCST) are frequently used since they enable well-controlled, narrow-size distributions and temperature-controlled binding to bacteria and free lectins [32,33,34,35,36,37,38,39,40,41]. Typically, these responsive microgels are synthesized using poly(*N*-isopropylacrylamide) (PNIPAM) with an LCST in the physiological range (32 °C) which is desired for potential biomedical applications [42]. However, polymers with acrylamide backbones such as PNIPAM have been shown to be genotoxic, mainly due to toxic degradation products, and are thus unlikely to be approved for in vivo applications [43,44]. As a more biofriendly alternative, monodisperse thermoresponsive polyethylene glycol (PEG)-based microgels have been developed that can be synthesized using straightforward free radical precipitation polymerization [45]. By copolymerizing di(ethylene glycol) methyl ether methacrylate (MEO_2_MA) and oligo(ethylene glycol) (OEGMA) at certain ratios toward P(MEO_2_MA-*co*-OEGMA) microgels with thermoresponsive behaviors similar to PNIPAM have been reported [46,47].

Based on these findings, this study explores the ability of carbohydrate-functionalized biocompatible P(MEO_2_MA-*co*-OEGMA) microgels to capture a carbohydrate binding lectin (ConA) and *E. coli*. A series of P(MEO_2_MA-*co*-OEGMA) microgels was prepared under variation of the crosslinker length and Man functionalization degree. The effect of these parameters on carbohydrate binding capabilities was determined by capturing ConA using microgel dispersions and observing the mobility of *E. coli* on thin microgel films (Figure 1).

## 2. Results and Discussion

### 2.1. Synthesis of Man/Gal-Functionalized P(MEO_2_MA-co-OEGMA) Microgels

The synthesis of the microgels was performed according to the route established by Cai et al. [45] using MeO_2_MA and OEGMA as monomers, sodium dodecyl sulfate (SDS) as a surfactant, ammonium persulfate (APS) as an initiator and different crosslinkers. As additional monomers, *N*-ethylacrylamide-α-d-mannopyranoside (ManEAm) and *N*-ethylacrylamide-α-d-galactopyranoside (GalEAm) were used to introduce Man/Gal units into the microgels similar to previously established carbohydrate-functionalized PNIPAM microgels (Figure 2) [32]. Varying the amount of ManEAm in the reaction solution allowed the control of the Man functionalization degree (Table 1). The Gal-functionalized microgels were used as a negative control sample since they do not bind to ConA or *E. coli*. A sample without carbohydrate monomers (PEG-EGDMA) was used as an additional non-binding control. To vary the microgel network density and the elastic modulus, crosslinkers of different length were used, where ethylene glycol dimethacrylate (EGDMA) represented the shortest and PEGDMA750 the longest crosslinker. The microgels are termed according to their carbohydrate functionalization degree (in µmol carbohydrate per g microgel) and the type of crosslinker used. For example, the microgel termed Man57-PEGDMA550 signifies a functionalization degree of 57 µmol Man per gram microgel and crosslinking by PEGDMA550 (Table 1).

The Man functionalization degrees for the microgels with different crosslinkers suggest that increased crosslinker lengths favor the integration of carbohydrate monomers into the microgel network. For precipitation polymerization methods toward microgels composed of LCST polymers, crosslinkers typically show a higher rate of network incorporation since they are bifunctional, leading to a higher crosslinking density in the microgel core. The incorporation of larger crosslinkers such as PEGDMA may be reduced by slower diffusion and screening of the reactive group due to coiling. This leads to a more homogeneous network structure [48,49], which is also more swollen due to the increased hydrophilicity of the long PEG-based crosslinkers. The resulting less dense network structure might improve the incorporation of the hydrophilic sugar units.

### 2.2. Morphology and Temperature-Dependent Swelling

Dynamic light scattering (DLS) was used to determine the hydrodynamic diameters (*D*_h_), the polydispersity, and the temperature-dependent swelling behavior of the microgels. The microgels with high sugar functionalization degree (Man135-EGDMA) showed the largest particle size (Figure 3), possibly due to the increased hydrophilicity of the mannose units leading to a higher swelling degree during the reaction. The increase of the crosslinker length (Man57-PEGDMA550, Man60-PEGDMA57) increased the microgel sizes. The DLS polydispersity indices of the microgels, as calculated from cumulant fits, were close to 0.1; therefore, the microgels can be considered near-monodisperse (Table 2) [50]. Compared to the unfunctionalized PEG sample (PEG-EGDMA), microgels with a higher sugar functionalization degree have a higher size polydispersity (PDI) as observed for carbohydrate-functionalized PNIPAM microgels [32]. The swelling ratios (*D*_h_ 20 °C/*D*_h_ 40 °C) were between 1.4 and 1.7 lower compared to sugar-functionalized PNIPAM microgels [32], which had a swelling degree above 2.0. The lower swelling degree of the MEO_2_MA-*co*-OEGMA microgels as observed earlier [45] could be caused by statistical effects owing to the random sequence of the comonomer backbone compared to the PNIPAM microgels, which are composed of homopolymer chains, thus showing a more defined phase behavior. The comparison of the swelling ratios suggests that the introduction of the longer PEG linkers reduces the temperature-dependent swelling/deswelling, as observed earlier by Lyon and coworkers [51].

Atomic force microscopy (AFM) images of dry microgel films on glass slides confirm the narrow size distribution and the increased sizes of microgels with the PEGDMA crosslinker (Figure 4a). In addition, the height profiles (Figure 4b) indicate a truncated sphere structure of the microgels. Microgels with the PEGDMA crosslinker attained a flat, pancake-like shape on the surface, i.e., the height of the microgels was only 15 nm for Man60-PEGDMA750, whereas microgels with short EGDMA attained heights around 50 nm. This suggests a decreased network density and a less dense microgel network owing to the longer crosslinkers [52].

### 2.3. Elastic Modulus

The elastic modulus (*E*) of polymer networks scales with the average mesh size ξ as *E* ~ *ξ*^−3^. Since *ξ*^−3^ is proportional to the crosslinker concentration elastic modulus measurements show the microgel crosslinker concentration and the effect of varying the crosslinker length. The elastic modulus was quantified in liquid by force indentation measurements at the apex of the microgels on solid support using sharp AFM tips as force probes [53,54].

As expected, microgels with the PEGDMA crosslinker showed lower elastic moduli (≈0.5 MPa) when compared to microgels with the shorter EGDMA crosslinker (2.7–3.7 MPa) (Figure 5). Studies on similar gels showed that the elastic modulus is reduced when using crosslinkers with a longer chain length [55]. The strong decrease in elastic modulus when using the PEGDMA linker could also be due to the increased hydrophilicity and slower diffusion compared to EGDMA during the precipitation reaction of the microgels resulting in a decreased incorporation into the microgel network. The differences in the elastic modulus for the EGDMA samples were within the standard deviation; a correlation with the carbohydrate functionalization degree or other compositional parameters apart from the crosslinker type was not observed. Overall, these results indicated a reduced network density of microgels crosslinked by PEGDMA, in line with AFM imaging of dried microgels (Figure 4).

### 2.4. ConA Uptake

Binding studies below the LCST (25 °C) between microgels and the lectin concanavalin A (ConA), which is a well-known model receptor for studying carbohydrate interactions [56], indicated specific binding to mannose-decorated microgels. Using this agglutination-based readout, we saw that ConA binding is not only affected by the carbohydrate density but also by the architecture of the microgels. To determine the binding affinity of each microgel with the lectin ConA, a quantitative ConA binding assay based on UV–VIS spectroscopy was performed [57]. The suspension of microgels was mixed with a solution of ConA. The tetrameric receptors formed aggregates with the microgels that were separated by centrifugation, and the residual ConA in the supernatant was then quantified by UV–VIS spectroscopy (Table 3). The results show that increased amounts of ConA were captured at increased Man functionalization degrees. On the other hand, more ConA per Man units in the microgels were captured at lower functionalization degrees, and the amount of bound ConA was several orders of magnitude smaller than the amount of sugar units in the microgels (Figure 6). This suggests that the majority of Man units in the microgel were not accessible for ConA binding. In addition, it is likely that the density of Man units was significantly larger when compared to the density of ConA binding sites, which show a minimum spacing of 7.1 nm on the ConA surface [58]. Interestingly, for longer crosslinkers, more ConA per Man units were captured, suggesting that increased crosslinker lengths reduced the network density and increased the network flexibility thereby increasing the accessibility of Man units in the microgel. Non-functionalized PEG-EGDMA microgels showed very low non-specific binding to ConA. Overall, the amount of captured ConA was similar when compared to Man-functionalized PNIPAM microgels at a similar functionalization degree [32].

### 2.5. E. coli Binding

We used type 1 fimbriated *E. coli* expressing the Man-receptor FimH to determine the ability of the microgels to capture carbohydrate-binding cells. Previous work on Man-functionalized PNIPAM microgels showed that they cluster *E. coli* in solution via Man binding [32]. Although we could show that the MEO_2_MA-*co*-OEGMA microgels captured ConA in solution under formation of clusters, they did not cluster *E. coli* below or above the LCST. Compared to the previously investigated PNIPAM microgels, MEO_2_MA-*co*-OEGMA microgels had a 3 times smaller diameter while having similar Man functionalization degrees. Therefore, due to their reduced size, the number of FimH-microgel interactions was smaller, which may lead to rapid unbinding, thereby preventing bacteria clustering.

To avoid microgel diffusion after unbinding from the bacteria and to maximize the microgel density, we studied the binding of *E. coli* to microgel surface coatings rather than dispersions. Therefore, densely packed monolayers of the microgels were prepared, and the mobility of the *E. coli* in the vicinity to the layers was assessed by fluorescence microscopy. From the microscopy images, it could be seen that surfaces coated with Man-presenting microgels permanently bound some of the bacteria, while unfunctionalized microgels or coatings with Gal functionalization showed faster bacteria movement and no permanent binding (Figure 7a). For a more quantitative analysis, time-lapse images were collected and pixel-wise grey value changes between consecutive frames were determined, where fast bacterial movement is characterized by strong positional gray value changes (see Appendix A). This analysis showed that all Man-presenting microgel surfaces achieved a similar reduction in bacterial movement compared to the Gal-presenting or unfunctionalized microgel coatings, indicating that the density of Man units was large in comparison to the density of FimH receptors on the bacteria. Therefore, an increase in Man density did not lead to an increase in *E. coli* binding. Overall, these results confirm the specific binding of *E. coli* to Man-functionalized microgel surfaces, which could be useful for capturing carbohydrate-binding pathogens.

## 3. Materials and Methods

### 3.1. Microgel Synthesis

All chemicals were obtained from Sigma-Aldrich (Darmstadt, Germany) and used without further purification if not stated otherwise. The copolymerization of MEO_2_MA, OEGMA, crosslinker, and sugar monomer was conducted in a 250 mL three-necked flask with a condenser. Then, 1.500 mL MEO_2_MA (8.1 mmol), 0.417 mL OEGMA (0.90 mmol, 500 g mol^−1^), the crosslinker and 10 mg SDS (0.033 mmol) were dissolved in 145 mL ultrapure water. As crosslinkers, either 0.0172 mL ethylene glycol dimethacrylate (EGDMA) (0.093 mmol), 0.0450 mL poly(ethylene glycol) dimethacrylate (PEGDMA) at a molecular height of 550 g mol^−1^ (PEGDMA550, 0.091 mmol) or 67.60 mg PEGDMA at a molecular height of 750 g mol^−1^ (PEGDMA750, 0.090 mmol) were used (Appendix A
Appendix A). The solution was stirred with a magnetic stir bar with 500 rpm, continuously flushing with nitrogen and heating to 70 °C. After 1 h, the deprotected monomer *N*-ethylacrylamide-α-d-mannopyranoside (ManEAm) or *N*-ethylacrylamide-α-d-galactopyranoside (GalEAm) was added (Appendix A
Appendix A), 200 mg (0.72 mmol) and 900 mg (3.2 mmol) were added for ManEAm, and 150 mg (0.54 mmol) was used in case of GalEAm. Next, the reaction was started by adding 57 mg ammonium persulfate (0.25 mmol), dissolved in 5 mL ultrapure water. After 6 h, the reaction was stopped in an ice bath. All microgels were purified by repeated centrifugation at 20,000 rpm.

### 3.2. Phenol Sulfuric Acid Method

As a colorimetric test to quantify the microgel carbohydrate functionalization degree, the phenol sulfuric acid method was used. A calibration curve with α-d-methylmannose was established by a dilution series of 500, 350, 250, 150, 50, 25 and 0 μM (Appendix A
Appendix A). To 500 μL of each α-d-methylmannose solution, 250 μL of a 5 mass % phenol solution and 1.5 mL sulfuric acid were added. For carbohydrate quantification of the microgels, 500 μL of a 5 mg mL^−1^ microgel solution was mixed with 250 μL of a 5 mass % phenol solution, and 1.5 mL 98% sulfuric acid. The mixtures were shaken for 30 min at room temperature. The absorbance was measured at 490 nm with UV–VIS spectroscopy (SPECORD 210 PLUS, Analytik Jena, Jena, Germany)

### 3.3. Dynamic Light Scattering

The measurements were conducted with a Malvern Zetasizer Nano ZS (Malvern Panalytical, Kassel, Germany) equipped with a He-Ne-laser (wavelength of 633 nm) as a light source. The scattered light was detected with a scattering angle of 173°. The microgel samples at a concentration of 0.1 mg mL^−1^ were prepared in a 1 cm polystyrene cuvette. These samples were measured in a temperature range of 14 °C to 54 °C at an increment of 2 °C and a 20 min equilibration time between each temperature step. The hydrodynamic diameters and cumulants were determined by the software provided by the manufacturer.

### 3.4. Atomic Force Microscopy (AFM)

Dry microgel films were imaged via AFM to determine the microgel morphology. The films were prepared by drying a 0.01 mg mL^−1^ microgel dispersion on glass coverslips followed by immersing in water and drying under a stream of nitrogen. For AFM imaging, the JPK NanoWizard 2 (JPK Instruments AG, Berlin, Germany) with cantilevers with a nominal spring constant of 40 N m^−1^ (HQ:XSC11/NO AL, MikroMash, Bulgaria) was used in tapping mode. For elastic modulus determination, a JPK NanoWizard 4 (Bruker Nano GmbH, Berlin, Germany) in the QI (quantitative imaging) mode, with cantilevers with a nominal spring constant of 2.7 N m^−1^ (HQ:XSC11/NO AL, MikroMash, Sofia, Bulgaria) with a nominal tip radius of 8 nm, was used. The measurements were conducted in lectin binding buffer (LBB, 10 mM HEPES, 1 mM CaCl_2_, 1 mM MnCl_2_, pH 7.4) using the microgel coatings as described above.

### 3.5. Quantitative ConA Binding Assay

A ConA concentration series at 1, 0.75, 0.5 and 0.25 mg mL^−1^ and microgel solutions at a concentration of 1 mg mL^−1^ were prepared. Then, 0.5 mL of the microgel solution and 0.5 mL of the ConA solutions were shaken for 24 h at 25 °C so that a precipitate would form. After a 30 min incubation, the microgels were separated by centrifugation and the ConA concentration of the supernatant was determined by UV–VIS spectroscopy at 280 nm. Each microgel-ConA solution was prepared in triplicate, and average values of the amount of captured ConA per 100 µg microgels was determined for the microgel concentration series.

### 3.6. Fluorescence Microscopy

The measurements were performed on an inverted fluorescent microscope (Olympus IX73, Hamburg, Germany) equipped with an Olympus UPlanFL N 60x/0.90 objective (Olympus, Hamburg, Germany) and CMOS camera (DMK 33UX174L, The Imaging Source, Bremen, Germany). The microgel surfaces were prepared on μ-Slides 18 Well-Flat Uncoated (ibidi, Martinsried, Germany) by drying 30 μL of 1 mg mL^−1^ microgel dispersion inside the μ-Slides overnight, followed by immersing the slides in water to remove excess microgels. Then, 30 μL of the *E. coli* solution in PBS (2 mg mL^−1^, OD600 = 0.4) was added and incubated for 30 min at 37 °C with 75 rpm. Image stacks with 30 images at an interval of 1 s were taken after the incubation time using µmanager [59]. The used bacteria strain was *E. coli* (PKL1162), which expresses the green fluorescent protein (GFP). To determine the mobility of the bacteria, the difference in the fluorescence intensity per pixel was compared between consecutive slides and averaged over the whole stack using the multi kymograph tool in FIJI [60]. To account for slightly varying numbers of bacteria for the different microgel surfaces, the data were normalized using the average stack gray values.

## 4. Conclusions

Taken together, a series of Man- and Gal-functionalized P(MEO_2_MA-*co*-OEGMA) microgels were synthesized varying the carbohydrate functionalization degree and crosslinker type. Depending on the crosslinker type, the microgels differed in their network density and elastic modulus where the longer PEGDMA-based crosslinker led to less dense and softer microgels. Decreasing the microgel network density resulted in increasing binding of ConA relative to the density of Man units in the network, presumably due to increased diffusion of ConA into the microgel network and increased accessibility of the Man units. While the take-up of ConA was overall comparable to previously established Man-presenting PNIPAM microgels, the binding of *E. coli* appeared to be reduced since no clustering of the microgels and *E. coli* was observed in solution. We suspect that this was due to the reduced size of the microgels compared to the PNIPAM microgels, enabling too few simultaneous FimH-Man complexes on a single microgel for permanent binding and microgel-bacteria cluster formation. On the other hand, surface coatings of the microgels showed that the mobility of *E. coli* in the vicinity of the coating was reduced due to specific binding. This suggests that biocompatible P(MEO_2_MA-*co*-OEGMA) microgels are generally suitable scaffolds to address carbohydrate-binding pathogens for potential biomedical applications. While ConA binding depended on the overall Man functionalization degree, the *E. coli* mobility on the microgels’ surfaces showed no dependence on the Man functionalization degree, which could be explained by the low density of FimH units on the bacteria compared to the Man density of the microgels. Future work will address the potential size dependence of bacteria clustering as well as the effect of temperature changes on the mobility of carbohydrate-binding cells on microgel coatings.

## Figures and Tables

**Figure 1 molecules-26-00263-f001:**
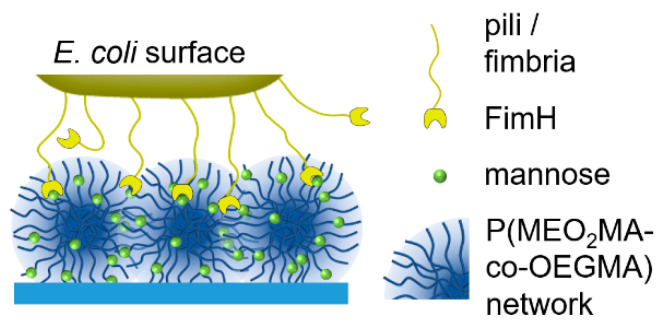
An illustration of *E. coli* binding to mannose-functionalized P(MEO_2_MA-*co*-OEGMA) microgel layers.

**Figure 2 molecules-26-00263-f002:**
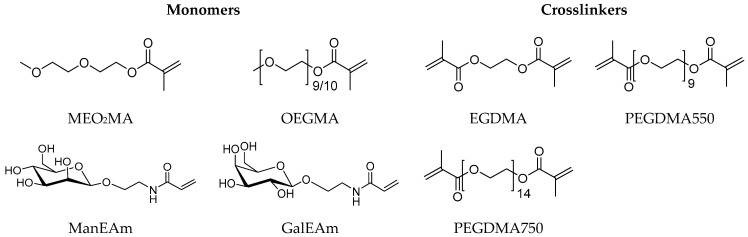
Monomers and crosslinkers used for the microgel synthesis.

**Figure 3 molecules-26-00263-f003:**
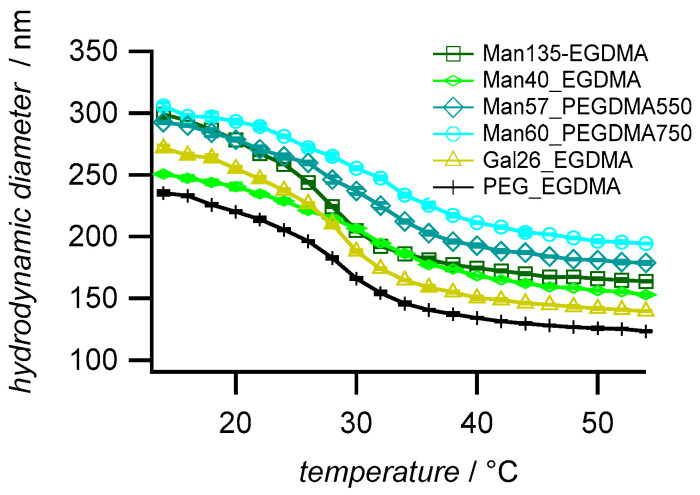
Temperature-dependent swelling behavior of the synthesized microgels. The values are averages (±SD) and were measured by dynamic light scattering in a temperature range of 14 °C to 54 °C with an increment of 2 °C and with a 20 min equilibration time between each temperature step.

**Figure 4 molecules-26-00263-f004:**
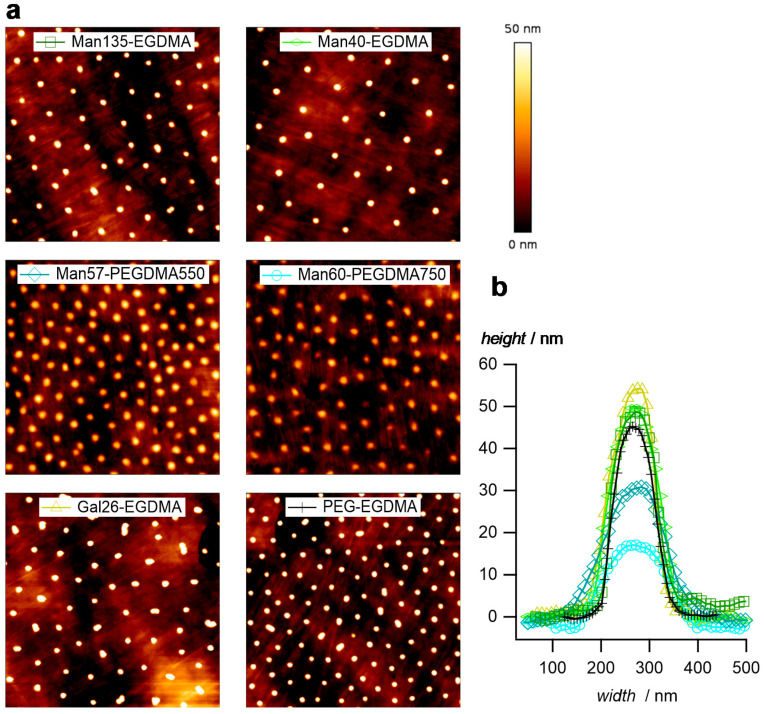
(**a**) Atomic force microscopy (AFM) images of the dried microgel films (image size 5 µm × 5 µm). (**b**) Typical profile plots selected microgels.

**Figure 5 molecules-26-00263-f005:**
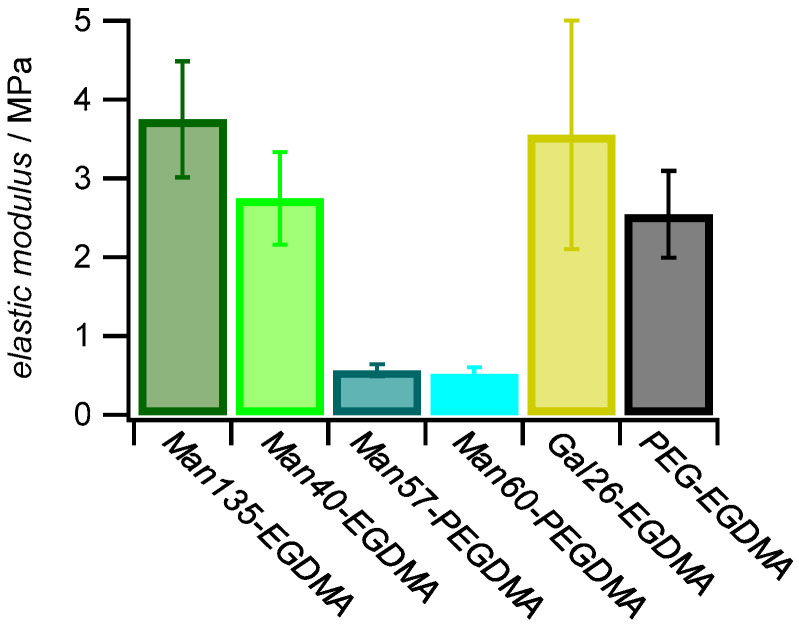
Effect of sugar functionalization and crosslinker on elastic modulus. The *E* (±SD) of 1 mg mL^−1^ dried microgel films was measured by AFM with a nominal tip radius of 8 nm.

**Figure 6 molecules-26-00263-f006:**
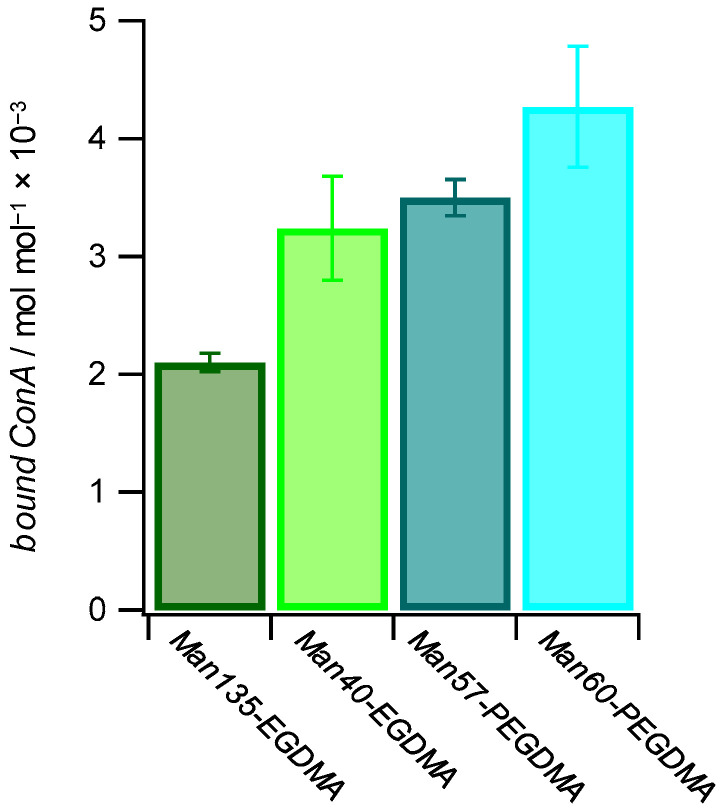
The amount of bound ConA increases with a higher mannose functionalization degree and with more flexible microgel networks. The values for bound ConA show how much ConA is bound by 1 mol of Man in the different Man-functionalized microgels.

**Figure 7 molecules-26-00263-f007:**
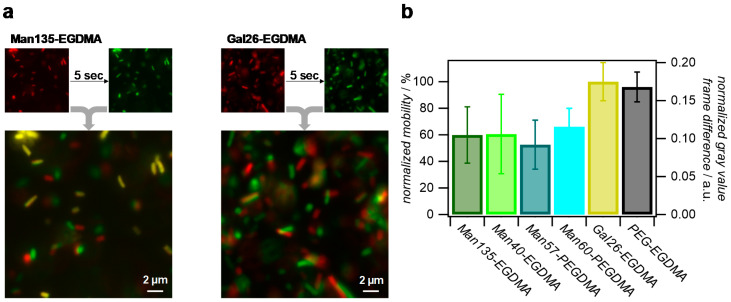
Analysis of the *E. coli* mobility on microgel surfaces via fluorescence microscopy of the green fluorescent protein (GFP)-tagged bacteria. (**a**) Comparison of *E. coli* positions on Man135-ethylene glycol dimethacrylate (Man135-EGDMA) and Gal26-EGDMA surfaces after a five-second delay. Yellow areas in overlay images signify no changes of positions and strongly impaired bacterial mobility due to strong binding. (**b**) Normalized gray value difference per pixel between frames in time-lapse images and normalized mobility using Gal26-EGDMA surfaces as a reference.

**Table 1 molecules-26-00263-t001:** Reactant compositions for microgel synthesis and carbohydrate functionalization degree ^a^.

Microgel Sample	Man/mmol	Gal/mmol	Crosslinker/mmol	Carbohydrate Reaction Ratio/% ^b^	Functionalization Degree/µmol g^−1 c^
Man135-EGDMA	3.2	-	0.093	35.5	135
Man40-EGDMA	0.72	-	0.093	8	40
Man57-PEGDMA550	0.72	-	0.091	8	57
Man60-PEGDMA750	0.72	-	0.090	8	60
Gal26-EGDMA	-	0.54	0.093	6	26
PEG-EGDMA	-	-	0.093	-	-

^a^ MEO_2_MA, oligo(ethylene glycol) (OEGMA), SDS and ammonium persulfate (APS) concentrations were constant (8.1 mmol, 0.90 mmol, 0.032 mmol and 0.25 mmol). ^b^ Carbohydrate concentration per MEO_2_MA+OEGMA concentration. ^c^ Man/Gal (µmol) per mass of microgel (g).

**Table 2 molecules-26-00263-t002:** Hydrodynamic diameter (*D*_h_), size polydispersity (PDI) and swelling degree—determined by DLS. The values represent averages over three measurements. The error represents the standard deviation.

Microgel Sample	Hydrodynamic Diameter *D*_h_ at 20 °C [nm]	PDI (DLS) at 20 °C	Swelling Ratio [*D*_h_ 20 °C/*D*_h_ 40 °C]
Man135-EGDMA	278 ± 0.8	0.115 ± 0.011	1.6
Man40-EGDMA	240 ± 2.6	0.069 ± 0.011	1.4
Man57-PEGDMA550	278 ± 1.1	0.092 ± 0.017	1.4
Man60-PEGDMA750	293 ± 1.8	0.149 ± 0.006	1.4
Gal26-EGDMA	255 ± 1.7	0.072 ± 0.021	1.7
PEG-EGDMA	220 ± 0.7	0.027 ± 0.012	1.6

**Table 3 molecules-26-00263-t003:** The quantity of ConA bound to 100 μg of microgels at 25 °C. The amount of ConA in microgel dispersion was fixed to 300 pmol tetrameric ConA.

Sample	Man Quantity [nmol]	Captured ConA [pmol]
Man135-EGDMA	13.5	44
Man40-EGDMA	4	20
Man57-PEGDMA	5.7	28
Man60-PEGDMA	6.0	38
PEG-EGDMA	0	1.8

## Data Availability

The data presented in this study are contained within this article and Appendix A.

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
