# Peer review of "Lectin and E. coli Binding to Carbohydrate-Functionalized Oligo(ethylene glycol)-Based Microgels: Effect of Elastic Modulus, Crosslinker and Carbohydrate Density"

_molecules, 2021, doi:10.3390/molecules26020263_

Round 1
Reviewer 1 Report
The way the results and analyses are described, it is hard to interest of this paper, what have we learned that is new? Biological relevance should be highlighted in the context of the many examples reported in literature, examples specific for conA and for FimH-bearing E. coli respectively, and not count solely on the very general reference, often without any direct relation, as is the case now. Some examples are given below:
microgels and lectins and bacteria (FimH):
DOI 10.1039/c7tb02964k Aqueous medium-induced micropore formation in plasma polymerized polystyrene: an effective route to inhibit bacteria adhesion
Polymers and lectins and bacteria (FimH):
DOI 10.1002/cnma.201500229: Affinity of Glycan-Modified Nanodiamonds towards Lectins and Uropathogenic Escherichia Coli
on mannose-Ethylene Glycol-linked inhibitors of FimH (Mana-triethylene glycol (6), Figure 2: best binder on micro glycan array) : DOI:10.1111/j.1365-2958.2006.05352.x The affinity of the FimH fimbrial adhesin is receptor-driven and quasi-independent of Escherichia coli pathotypes:
(conA) Figure 3. Comparison of ConA binding to Man6 and Man-R. (a) ConA binding at high neoglycoprotein density (1:0); shown are pairs of spots for Man-R and Man6 at a single ConA concentration (189 nM) and binding curves over a range of ConA concentrations. (b) ConA binding at low neoglycoprotein density (1:7); DOI 10.1021/ja100608w J. AM. CHEM. SOC. 9 VOL. 132, NO. 28, 2010 An Array-Based Method To Identify Multivalent Inhibitors
Why are the lectins called receptors, this is opposite to reality?
Author Response
Reviewer 1
The way the results and analyses are described, it is hard to interest of this paper, what have we learned that is new? Biological relevance should be highlighted in the context of the many examples reported in literature, examples specific for conA and for FimH-bearing E. coli respectively, and not count solely on the very general reference, often without any direct relation, as is the case now. Some examples are given below:
microgels and lectins and bacteria (FimH):
DOI 10.1039/c7tb02964k Aqueous medium-induced micropore formation in plasma polymerized polystyrene: an effective route to inhibit bacteria adhesion
Polymers and lectins and bacteria (FimH):
DOI 10.1002/cnma.201500229: Affinity of Glycan-Modified Nanodiamonds towards Lectins and Uropathogenic Escherichia Coli
on mannose-Ethylene Glycol-linked inhibitors of FimH (Mana-triethylene glycol (6), Figure 2: best binder on micro glycan array) : DOI:10.1111/j.1365-2958.2006.05352.x The affinity of the FimH fimbrial adhesin is receptor-driven and quasi-independent of Escherichia coli pathotypes:
(conA) Figure 3. Comparison of ConA binding to Man6 and Man-R. (a) ConA binding at high neoglycoprotein density (1:0); shown are pairs of spots for Man-R and Man6 at a single ConA concentration (189 nM) and binding curves over a range of ConA concentrations. (b) ConA binding at low neoglycoprotein density (1:7); DOI 10.1021/ja100608w J. AM. CHEM. SOC. 9 VOL. 132, NO. 28, 2010 An Array-Based Method To Identify Multivalent Inhibitors
We feel that we sufficiently explained the relevance of our study but perhaps we should again highlight the relevance of microgels as a new class of carbohydrate presenting scaffolds (see changes below). To show the relevance of our work, these are the key points of the paper:
- Carbohydrate functionalized microgels as synthesized in this work are a quite recent and valuable tool to specifically bind carbohydrate binding cells and control their interaction via external stimulus, see references 26-30 (articles from 2019 or newer)
- We show for the first time how biocompatible PEG-based microgels interact with lectins and E. coli.
- We study the effect of the microgels elastic modulus and network density on lectin binding. Studying the effect of these material parameters is often overlooked but essential to understand the biomolecular interactions.
- We show for the first time how E. coli interact with mannose functionalized PEG-based microgels and compare these results to previously investigated PNIPAM microgels.
We also feel that discussing the articles suggested by the reviewer are helpful but not critical. The article (10.1002/cnma.201500229) does not present glycan functionalized materials or carbohydrate binding. The second article (10.1002/cnma.201500229) does not discuss glycan functionalized polymer gels but an entirely different class of glycan scaffold. The next article (10.1111/j.1365-2958.2006.05352.x) is a very old paper on the FimH receptor (we have covered this in more recent overview articles, see references 5 and 6). The article 10.1021/ja100608w presents glycan arrays to identify carbohydrate binding proteins, which is again a different topic.
Nevertheless, we feel that the use of microgels should be motivated a bit more. Therefore, we add the following text (page 2 line 5): “In addition to their straightforward synthesis, microgels allow for preparing robust surface coatings by simple physisorption methods, e.g. via drop-casting, spin coating or dip-coating.[31] Carbohydrate functionalized microgels are highly hydrated and soft, thus mimicking properties of the extracellular matrix or glycocalyx, which sets them apart from other glycan presenting scaffolds.”
Why are the lectins called receptors, this is opposite to reality?
Lectins can be called receptors since they bind carbohydrates as per definition, i.e. they are carbohydrate receptors.
Reviewer 2 Report
Authors demonstrate a synthesis of carbohydrate-functionalized biocompatible poly(oligo(ethylene glycol) methacrylate microgels. The ratio behind the monomers chosen and the copolymer is biocompatibility. Authors analyzed prepared products mainly by dynamic light scattering, and give evidence of nearly monodisperse hydrogel product particle sizes. They also demonstrated that decoration with Man or Gal (or none) can change important biological properties. Namely, binding to ConA or E. coli. The paper is in my view important peace to the knowledge of functional materials and their chemical and biological behavior/properties.
Some broad and specific comments are highlighted or are in notes directly in the attached pdf of the manuscript. They are of minor importance, however, I do recommend carefully pore through. It can improve the paper quality and professional/scientific shape.

Author Response
We applied all formatting changes as suggested by reviewer 1, except for the following two comments (taken from the commented PDF):
Comment regarding table 3:
Use:
Man/mmol
Gal/mmol
crosslinker/mmol
carb. .../% - however it can be ommited, it is obvious from reaction composition (3,2/0,093 etc.)
degree of functionalisation/umol g-1 (umol of what an g of what is allready defined)
We applied the suggested changes except for omitting the “carbohydrate reaction ratio / % b “ column. Although redundant, we feel that this helps understanding the differences in the microgel composition.
Comment regarding figure 3, figure 4 and figure 5:
The name of varieble should be mentioned in the text or in the Fig. description, not here.Here use d (italics) as diamater or just diameter
We feel that although using variables as axis labels is valid, using the full names as axis labels helps clarity and readability. This is a chemical publication after all and the variables are not used extensively in the text.
Round 2
Reviewer 1 Report
The authors have not improved the manuscript, the relevance of their study is still absent and certainly has not been placed within the context of current state of the art. Therefore I remain with my opinion that the paper does not present any, absolutely no, novel insight and should be rejected.